



# Small river plumes off the north-eastern coast of the Black Sea under average climatic and flooding discharge conditions

Alexander Osadchiev[1], Evgeniya Korshenko[2]

[1]Shirshov Oceanology Institute, Moscow, Russia
[2]Zubov State Oceanographic Institute, Moscow, Russia

*Correspondence to*: Alexander Osadchiev (osadchiev@ocean.ru)

**Abstract.** This study is focused on the impact of discharge of small rivers on delivery and fate of fluvial water and suspended matter at the north-eastern part of the Black Sea under different local precipitation conditions. Several dozens of mountainous rivers inflow into the sea at the study region and most of them, except the several largest, have small annual runoff and limitedly affect adjacent coastal waters under average climatic conditions. However, discharges of these small rivers are characterized by quick response to precipitation events and can dramatically increase during and shortly after heavy rains which are frequent in the considered area. Delivery and fate of fluvial water and terrigenous sediments at the study region under average climatic and rain-induced flooding conditions were explored and compared using in situ data, satellite imagery and numerical modelling. It was shown that the point-source spread of continental discharge dominated by several large rivers during average climatic conditions can change to the line-source discharge from numerous small rivers situated along the coast in response to heavy rains. Intense line-source runoff of water and suspended sediments forms a geostrophic alongshore current of turbid and freshened water, which induces intense transport of suspended and dissolved constituents discharged with river waters in a north-western direction. This process significantly influences water quality and causes active sediment load at large segments of narrow shelf at the north-eastern part of the Black Sea as compared to average climatic discharge conditions.

**Key words:** river plume, river discharge, small rivers, coastal water quality, terrigenous sediments, rain-induced flood, Black Sea

## 1 Introduction

Continental discharge is one of the main sources of terrigenous sediments, nutrients and anthropogenic pollution at the sea, and it can significantly affect seabed morphology, water quality, primary productivity and fishery in coastal areas (e.g., Emmet, 2006; Milliman et al., 2007; Zhou et al., 2008; Rabalais, 2010). Generally the majority of fluvial runoff and the related discharge of its suspended and dissolved constituents on a regional scale are provided by the largest local rivers, while small rivers, i.e., rivers with small drainage basins and small annual discharge, usually play an insignificant role. Moreover, nowadays most of the world small rivers are not covered by regular hydrological and discharge measurements



which cause lack of information about their runoff volume and variability (Vorosmarty et al., 2001; Hrachowitz et al., 2013). Thus, the studies focused on delivery and fate of river-borne dissolved and suspended matter at coastal zones generally consider only one or several largest rivers of a study area, while the influence of small local rivers is neglected.

However, under certain terrain and climatic conditions cumulative discharge from small rivers can greatly increase in response to heavy rains and become comparable or even exceed runoff of large rivers (Mertes and Warrick, 2001; Wheatcroft et al., 2010; Kniskern et al., 2011; Saldias et al., 2016). This rain-induced flooding discharge of small rivers can significantly influence land-ocean fluxes of fluvial water, sediments, nutrients and pollutants as well as modify structure and intensity of coastal transport pathways for certain world regions at least on a short-term scale, as addressed in a number of relevant studies (e.g., Milliman and Syvitski, 1992; Meybeck et al., 2003; Brodie et al., 2010; Hilton et al., 2011; Bao et al., 2015; Warrick and Farnsworth, 2016).

This article is focused on the impact of discharges of small rivers on delivery and fate of fluvial water and suspended sediments at the north-eastern coast of the Black Sea under different discharge conditions. We considered two trial periods in summer and autumn (25 May – 4 July 2011 and 6-19 September 2011), characterized by seasonal freshet and draught discharge conditions. Several flash flooding events were registered during both periods which influenced large segments (50-200 km long) of the coast.

Based on in situ data, satellite imagery, and numerical modelling, we reconstructed the daily volumes of fluvial water and terrigenous sediments discharged during the trial periods from 20 biggest rivers of the study region using a recently developed method described in Osadchiev (2015). Then using a nested combination of the Eulerian model INMOM and the Lagrangian model STRiPE we simulated spread of the buoyant plumes generated by these rivers during the trial periods in two modes characterized by different river discharge conditions. The first mode was run using the reconstructed discharge data ("real" mode), while the second mode used smoothed discharge data; therefore, flash floods were substituted by periods of average seasonal discharge ("averaged" mode). Also we simulated transport and settling of river-borne terrigenous sediments discharged during the first trial periods for both discharge modes. Based on the obtained results of the numerical modelling, we reconstructed the transport patterns of river-borne suspended sediments for normal and flash flooding discharge conditions and showed their significant difference.

The article is organized as follows. Section 2 provides detailed information about the study region. Satellite and in situ data collected in the study region and used for model application and validation are described in Section 3. Description of the buoyant river plumes formed at the study area as well as reconstruction of daily hydrographs of the 20 largest rivers of the study area are given in Section 4. Section 5 is focused on the general description of the numerical model used to reproduce delivery and fate of fluvial water and river-borne terrigenous sediments. The results of numerical simulations of river discharge spread and transport of suspended sediments under average seasonal and flooding discharge conditions as well as the related discussion are given in Section 6. A brief summary and the conclusions are presented in Section 7.



## 2 Study area

### 2.1 Topography and precipitation conditions

The study region includes a 300-km section of the Russian coast of the Black Sea (RCBS) between the city of Novorossiysk

and the city of Sochi (Fig. 1). Drainage basin of RCBS is a narrow area limited by the Greater Caucasus Range (GCR) at the

east and the sea coast at the west. Height of GCR increases from 400 m in the north to 3200 m in the south, while the

distance between GCR and the sea coast grows from 10 km to 40 km.

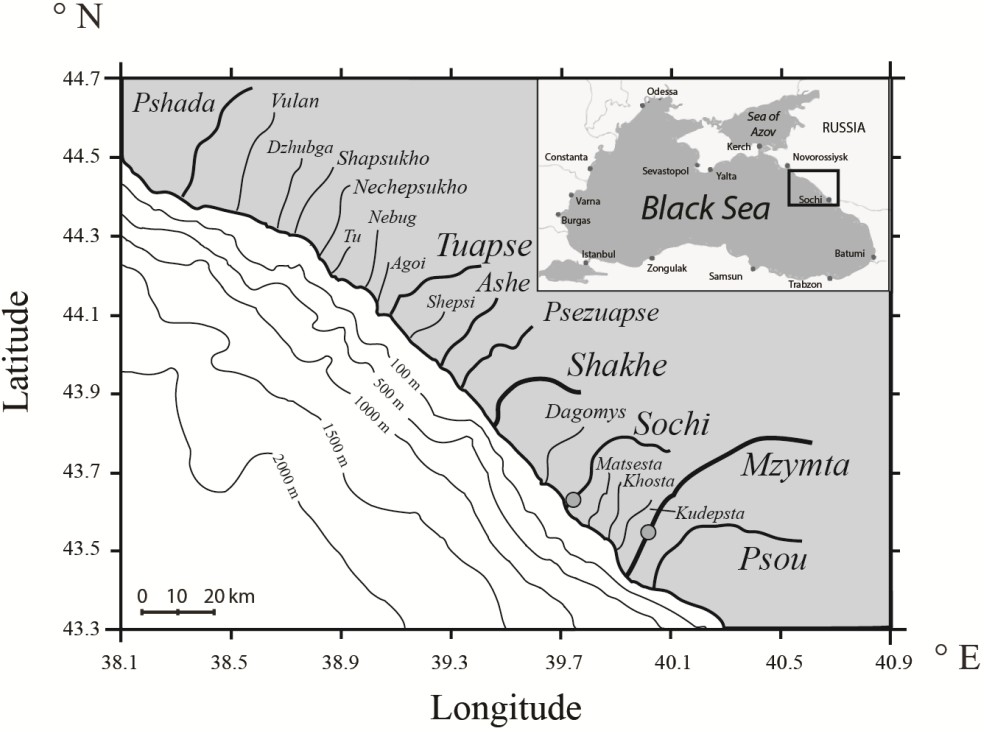

**Figure 1: Location of the study region, deposition of the 20 largest rivers, gauge stations (grey circles at the Mzymta and Sochi**
**rivers), and bathymetry of the coastal area.**

According to the Koppen climate classification, the study area lies in a humid subtropical zone (Cfa) with hot and humid

summers and cold winters. Complex topography of RCBS greatly influences the local atmospheric circulation and the local

precipitation regime. In particular, orographic lift induces frequent and intense rains at the southern side of the GCR. Mean

annual precipitation volume steadily grows (following the increase of height of GCR) from 700 mm at the northern part of

RCBS to 1700 mm at the southern part. The local precipitation is characterized by a significant seasonal variability with a

maximum in winter and a minimum in summer. However, intense precipitation events can occur in all months of the year, in

particular, the maximal amount of daily rainfall at the study region (298 mm) was registered in June.



## 2.2 River discharge

The steep gorges located between the numerous spurs of GCR form the drainage basins of more than 50 rivers and watercourses, which inflow into the sea at the study region. The areas of these basins are relatively small and only eight of the considered rivers have annual discharge greater than 10 m$^3$ s$^{-1}$. The total annual continental runoff from RCBS to the sea

is estimated as 7 km$^3$ (Jaoshvili, 2002).

Discharge of the Mzymta River, which is the biggest river of the study area, is characterized by a drought period in autumn and winter and a freshet in spring and early summer associated with snow melting. The other rivers of RCBS are mainly rain-fed and their annual runoff volume is formed mainly during short-term floods (15-25 annually) with sharp rises and falls of discharge caused by the following reasons. The steep slopes of the drainage basins (up to 40-60º) of these rivers, their

small sizes (below 900 km$^2$) and high drainage density (0.85-1.05) cause quick delivery of rain water into the river channels. As a result discharge of these rivers to the sea can dramatically increase during several hours in response to an active precipitation event (Balabanov et al., 2011; Alexeevsky et al., 2016). For example, the heavy rain on 31 July – 1 August 1991 caused increase of discharge of the Tuapse River from 0.8 to 2300 m$^3$ s$^{-1}$ during less than 5 hours, while the subsequent recession of its runoff lasted for 3 days. Flooding periods are mainly registered during winter (November-March), which

provides 65-80% of total annual discharge of the small rivers of RCBS.

According to Jaoshvili (2002) approximately 106 m$^3$ of sediments are discharged from the RCBS rivers to the north-eastern part of the Black Sea. Coarse sediments, which constitute approximately 1/3 of the total sediment volume, are deposited at the shallow areas near the shore, while fine sediments are mainly transported offshore and settle to the deep ocean. Annual average suspended sediment concentrations in the rivers of the study area is 50-150 g m$^{-3}$; however, their daily and seasonal

values are characterized by significant variability. In particular, average seasonal suspended sediment concentration in the Mzymta River varies between 48 g m$^{-3}$ in February and 226 g m$^{-3}$ in May, while daily sediment concentrations are spanned between 0 and 11000 g m$^{-3}$. On average, concentration of suspended sediments in the Mzymta River exceeds 1000 g m$^{-3}$ during 6 days in a year (Balabanov et al., 2011).

## 2.3 Coastal circulation and bathymetry

The bathymetry of the study region is characterized by the narrow shelf, the distance from the shore to the 100-m isobath varies between 2 and 15 km. Further offshore the steep continental slope descends to a depth of 1000 m at a distance of 20-30 km from the shore (Fig. 1). Multiple underwater canyons are located at the continental shelf and continental slope of the study region. The narrowest shelf is located near the Pshada, Shakhe, Mzymta, and Psou river estuaries, and these areas are characterised by intense shelf erosion.

General water transport of the Black Sea is governed by a current system cyclonically circulating along the continental slope which is generally referred as the Black Sea Rim Current (RC) (e.g., Oguz et al., 1992; 1993; Kortotaev et al., 2003). Velocity of RC at the north-eastern part of the Black Sea is 0.2-0.5 m s$^{-1}$. Coastal circulation in the study region is also



influenced by nearshore anticyclonic eddies (NAE), which are regularly formed between the main flow of RC and the coast due to baroclinic instability caused by wind forcing and coastal topography. Diameters of the majority of NAE do not exceed 60 km; however, spatial scales of the biggest NAE can be up to 160 km. NAE formed in the study region moves in a north-western direction along the continental slope with average velocity of 0.02-0.04 m s$^{-1}$, while their average rotating velocity is

0.05-0.4 m s$^{-1}$ (Ginzburg et al., 2002; Zatsepin et al., 2003; Kubryakov and Stanichny, 2015a; 2015b).

## 3 Data

The in situ measurements used in this study were performed during two field surveys organized by Shirshov Oceanology Institute on 28-30 May 2011 and 15-19 May 2012 at the southern part of RCBS (Fig. 1). Field works took place in the coastal areas influenced by the Dagomys, Sochi, Matsesta, Khosta, Kudepsta, Mzymta, and Psou rivers. Field surveys

included continuous measurements of salinity and concentrations of total suspended matter (TSM), coloured dissolved organic matter (CDOM), and chlorophyll a (Chl-a) at the surface layer at the areas adjacent to the river estuaries. The measurements were performed along the ship track using a ship-mounted pump-through system equipped by a CTD instrument (*SeaBird SBE911*) and the Ultraviolet Fluorescent LiDAR and were organized as cross-shore transects, the distances between neighbouring transects in proximity of the river estuaries were 100-150 m. Additionally vertical profiles

of salinity were measured within the river plumes using a CTD instrument (*SeaBird SBE19plus*). The explicit description of the LiDAR instrument and the algorithms used for retrieving TSM, CDOM, and Chl-a are given in Palmer et al. (2013). The detailed information about the field surveys and the measurements described above can be found in Zavialov et al. (2014).

Satellite data used in this study included EnviSat MERIS L1 satellite imagery provided by the European Space Agency (ESA) and L4 sea surface temperature (SST) products derived and distributed in the framework of the Copernicus Marine

Environment Monitoring Service (CMEMS) project. EnviSat MERIS L1 satellite products with 300 m spatial resolution were used for retrieving maps of sea surface distributions of TSM, CDOM, and Chl-a using MERIS Case-2 Regional water processing module (Doerffer and Schiller, 2008). The resulting TSM and Chl-a distributions were validated against the results of analysis of the water samples collected at the surface layer and optical remote sensing performed by LiDAR at the southern part of the study region. Daily gap-free SST maps of the Black Sea with 0.0625° spatial resolution were obtained

from processing and statistical interpolation of night-time measurements collected by the infrared satellite sensors mounted on different satellite platforms. The details of SST data processing are described in Nardelli (2009; 2013).

Meteorological data for the Black Sea area were calculated by the Weather Research and Forecasting (WRF) model (version 3.6) using the Lambert conformal conic projection with horizontal spatial resolution of 10 km. The vertical coordinate was represented by 35 levels and the time step was set equal to 90 s. Initial and boundary conditions for WRF were obtained

from the National Centers for Environmental Prediction (NCEP) Final Analysis product. Finally, in this work we used daily gauge data of river discharge measured at the Sochi and Mzymta rivers (Fig. 1) and provided by the Federal Service for Hydrometeorology and Environmental Monitoring of Russia.



The in situ, satellite and WRF model data described above were used, first, for evaluation of the river discharge for the ungauged rivers of the study region (Section 4.2), and, second, for application and validation of the numerical model, which reproduced delivery and fate of fluvial water and river-borne terrigenous sediments at the north-western part of the Black Sea during the simulation periods (Section 6). In particular, the ability of the numerical model to reproduce submesoscale variability of the river plumes of RCBS, which is crucial for this study, was proved based on the salinity and TSM in situ data collected during 28-30 May 2011 at the field cruise (Section 6.1).

## 4 River plumes at RCBS

### 4.1 Structure and variability of river plumes

The multiple rivers of RCBS form buoyant plumes adjacent to the river estuaries. Under the average climatic discharge conditions the most extensive river plume is generated by the Mzymta River, the largest river of the study area, which mean monthly discharge varies between 20 m$^3$ s$^{-1}$ and 120 m$^3$ s$^{-1}$.

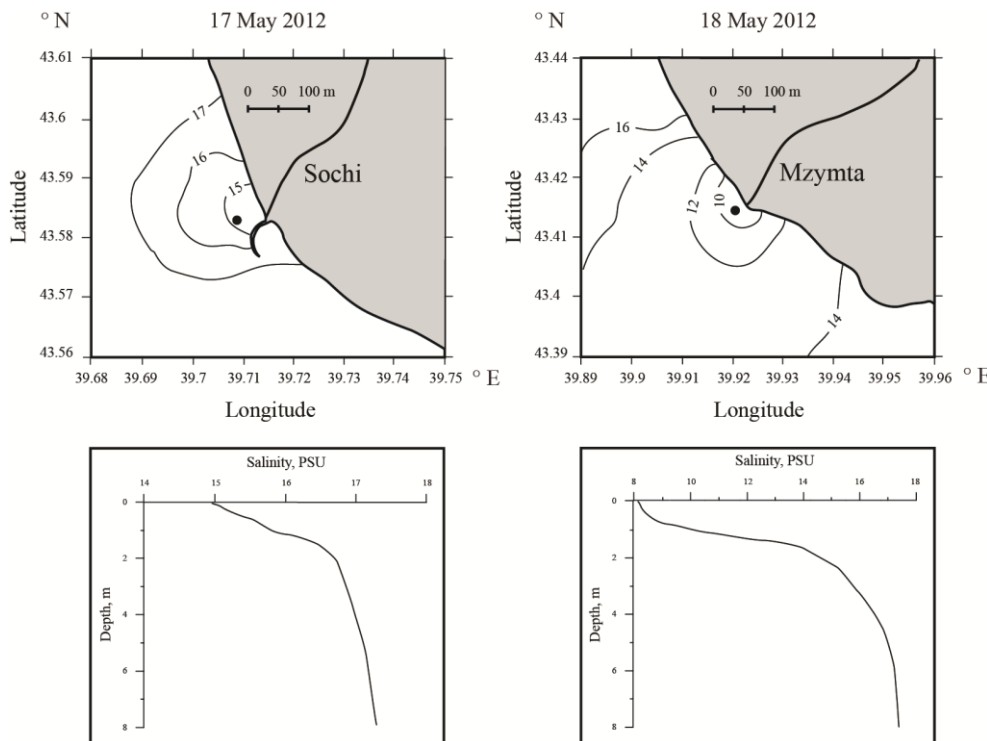

**Figure 2: Surface salinity distributions (top) and vertical salinity profiles (bottom) at the Sochi plume on 17 May 2012 (left) and the Mzymta plume on 18 May 2012 (right).**




Generally the Mzymta plume occupies less than 10 km² of the coastal zone, its depth does not exceed 5 m; however, during spring and summer freshet periods it can increase up to 50 km². The horizontal and vertical spatial scales of the other river plumes of RCBS are even smaller except short-term periods of flooding discharge induced by heavy rains. An example of surface salinity distributions and vertical salinity profiles near a river estuary for small (Sochi, 16 m³ s⁻¹, 17 May 2012) and

large (Mzymta, 81 m³ s⁻¹, 18 May 2012) river plumes are shown in Fig. 2.

The rivers of the study region are characterized by elevated concentrations of suspended sediments as compared to sea water. As a result, salinity correlates well with turbidity at the surface layer and the river plumes can be effectively detected at optical satellite imagery, as addressed in Zavialov et al. (2014) and Osadchiev (2015) (Fig. 3). Thus, ocean color remote sensing is an efficient tool for monitoring river plumes at the study region.

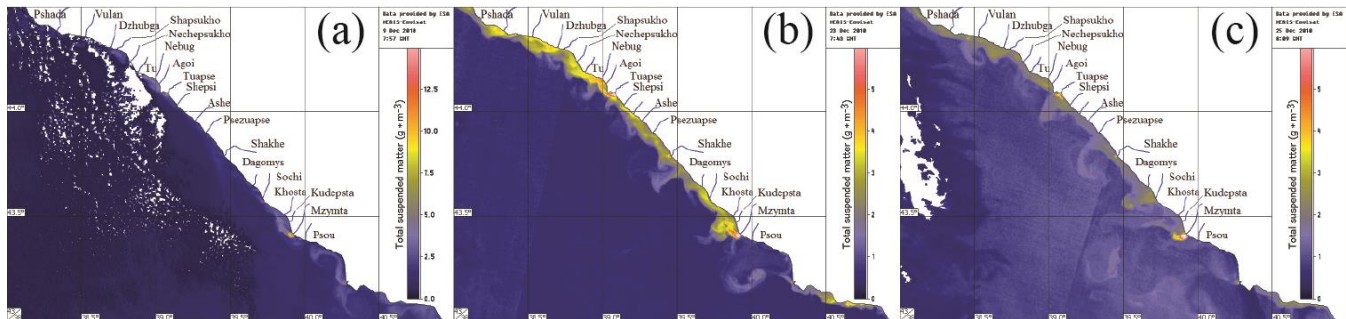

**Figure 3: Satellite derived surface TSM distribution at RCBS before (a), during (b), and after (c) a flash flooding event.**

Both in situ and satellite measurements revealed high spatial and temporal variability of the river plumes of RCBS. Their
area, form, and shape can significantly change during less than 1 day in response to variations of river discharge and local wind forcing (Korotkina et al., 2011; 2014; Zavailov et al., 2014). Under the average discharge conditions the river plumes are distinctly separated because their spatial scales do not exceed the distances between the river estuaries. However, during rain-induced floods the areas of the river plumes dramatically increase and they can collide and coalesce with neighbouring plumes. The most intense precipitation events cause formation of a continuous coastal stripe of turbid water which can be
observed at satellite imagery (Fig. 3b). After the end of a flooding period this stripe dissipates and river plume areas decrease to their average seasonal sizes (Fig. 3c).

## 4.2 Reconstruction of river discharge

In this work we explored discharge of the 20 largest rivers of the study region. The list of them sorted from north to south is as follows: Pshada, Vulan, Dzhubga, Shapsukho, Nechepsukho, Tu, Nebug, Agoi, Tuapse, Shepsi, Ashe, Psezuapse, Shakhe,
Dagomys, Sochi, Matsesta, Khosta, Kudepsta, Mzymta and Psou. These rivers are estimated to provide approximately 95% of the annual runoff of fluvial water and sediments from RCBS (Jaoshvili. 2002).



The Mzymta and Sochi rivers are the only rivers of the study region with available daily gauge measurements during the trial periods (25 May – 4 July 2011 and 6-19 September 2011). Discharge volumes of the other 18 rivers were evaluated using the method described in Osadchiev (2015). The general idea of this method is to use satellite-derived properties of a river plume for reconstruction of conditions of its formation, in particular, river discharge volume. First, the spatial extent and the shape

of the river plume are identified at satellite imagery. Second, a hydrodynamic model, which simulates formation of river plumes, is run with a variety of forcing conditions to identify the discharge rate that provides the best match between modelled and observed plumes.

The numerical model, based on the method described above, was tuned for the Mzymta and Sochi rivers and validated against gauge data. This model analysed TSM, CDOM, and Chl-a distribution maps derived from EnviSat MERIS satellite

imagery to identify river plumes at the study area. Then it applied the STRiPE model for simulating formation of river plumes. The details of the method, the numerical model, and its validation are given in Osadchiev (2015). After passing the validation procedure the model was applied to quantify discharge of the ungauged rivers of RCBS during the trial periods. The discharge values for the considered rivers were obtained for 11-13 out of 41 days of the first trial period and 6-8 out of 14 days of the second period. The number of days with reconstructed discharge is not the same for the considered rivers,

because it depends on feasibility of identification of river plumes, i.e., availability of cloud-free satellite imagery at the respective coastal areas.

However, discharge values during the days of the trial periods, which were not covered by appropriate satellite images, remained unknown. They were prescribed based on the reconstructed discharge data and WRF precipitation data in the following manner. The satellite-derived discharge values revealed relatively uniform river runoff during the trial periods

except several short-term flooding events with significantly elevated discharge. The flash floods at the rivers showed good correlation with the rain events at the respective river basins reconstructed by the WRF model. These rain events were observed during 25-27 May, 30 May, 21-23 June, and 27-30 June at the first trial period and during 9-12 September at the second trial period. Based on this fact, we presumed that the periods of peak discharge during the flooding events correspond to the periods of active precipitation registered at the respective river basin. Thus, the satellite-derived discharge values were

linearly extrapolated till the day of formation of a flooding event, followed by sharp increase of discharge to the peak value and linear decrease to the average discharge conditions after the end of a flooding event.

The resulting hydrographs of the Shakhe, Sochi, Mzymta, and Psou rivers, which annual runoff is greater than 15 m$^3$ s$^{-1}$ and which are, hereinafter, referred as the large rivers of RCBS, are presented in Fig. 4 together with the variation of the total discharge of the other 15 rivers, hereinafter, referred as the small rivers of RCBS. The measured and reconstructed

discharges of fluvial water from the 20 biggest rivers of RCBS show that continental runoff is dominated by the several biggest rivers of the study region during the periods of average seasonal discharge. The Shakhe, Sochi, Mzymta, and Psou rivers provided 65-85% of daily continental runoff during the trial periods. Flooding events significantly change this proportion, especially during autumn draught, when the average seasonal discharge rates of all rivers of the study area are



relatively low. In particular, the share of the large rivers of RCBS in total runoff during the flash floods decreased to 45% during the first trial period and to 20% during the second trial period.

After evaluation of the daily river discharge values we reconstructed the daily concentrations of suspended sediments at the near-field parts of the considered river plumes, which were used as input date for the numerical modelling addressed in

Section 6. First, the TSM concentrations were retrieved from the satellite data for the respective days of the trial periods. They showed strong logarithmic dependence on the reconstructed river discharge values separately within the large rivers and the small rivers of RCBS. The Pearson correlation coefficient between the $\ln(C)$ and $\ln(Q)$ was equal to 0.82 for the large and 0.8 for the small rivers, where $C$ is the TSM concentration in g m$^{-3}$, $Q$ is the river discharge in m$^3$ s$^{-1}$. Based on this fact, we used the obtained equations $C = Q^{1.2}$ for large rivers and $C = 9Q^{0.9}$ for small rivers, which are consistent with the

previous studies (Jaoshvili, 1986; Balabanov et al. 2011), for reconstructing the daily variability of the TSM concentrations of the rivers of RCBS during the trial periods.

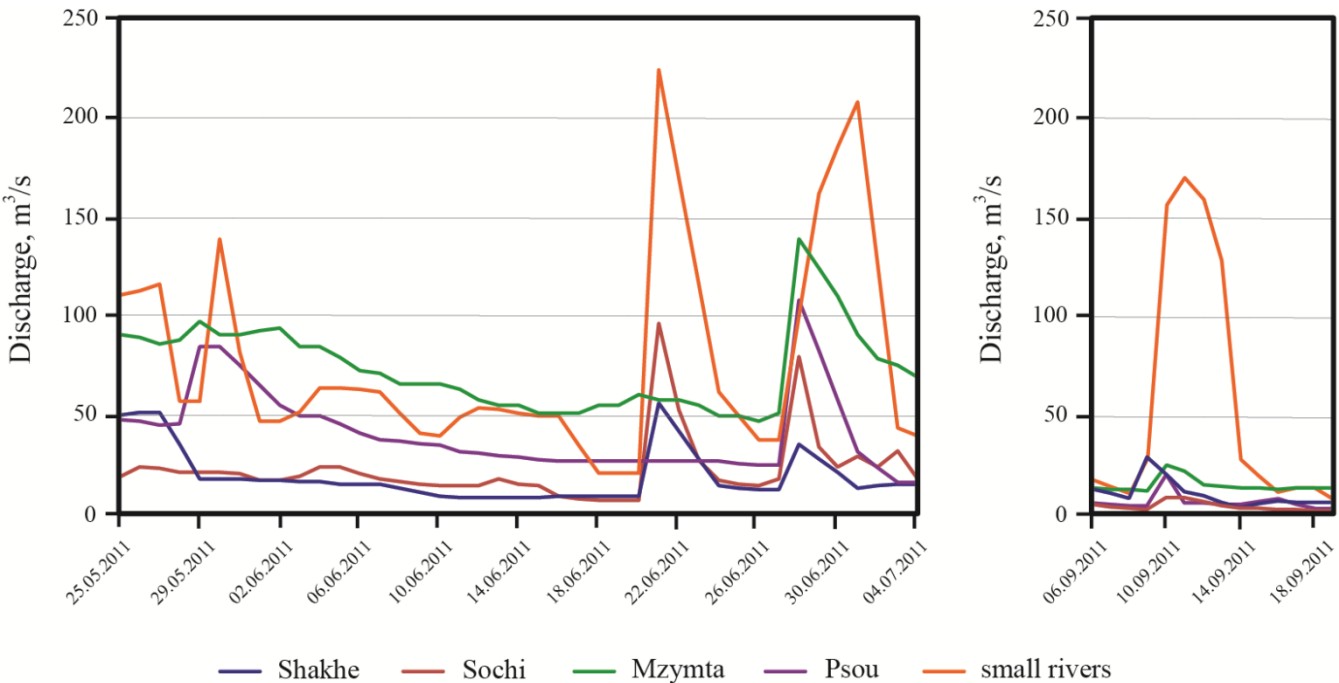

**Figure 4: Daily discharge rates of the rivers of RCBS during 25 May – 4 July 2011 (left) and 6-19 September 2011 (right)**
**measured at the gauge stations (for the Mzymta and Sochi rivers) and reconstructed using satellite imagery and numerical modelling (for the Shakhe, Psou and the small rivers).**

## 5 Model

In this work we applied numerical modelling to study delivery and fate of fluvial water and terrigenous sediments at RCBS under average seasonal and flooding river discharge conditions. Accurate simulation of the sub-mesoscale river plume



dynamics is essential for correct reconstruction of transport, mixing, and settling of fluvial water and suspended sediments at the coastal area. On the other hand, correct simulation of the mesoscale coastal circulation is also important, because, first, it influences river plume dynamics and, second, it governs transport and settling of river-borne suspended sediments after they sink beneath the plumes into the ambient ocean. For this reason we used a nested combination of Eulerian and Lagrangian

numerical models, which accurately reproduce both the mesoscale sea circulation and the sub-mesoscale dynamics of the multiple river plumes of the study area.

River-borne terrigenous sediments are tracked as passive tracers of river outflow. Initially tracers are transported by buoyant plume waters which dynamics is simulated by the STRiPE module. After the sediment particle settles beneath the plume its movement is governed by the ambient coastal circulation, reproduced by the INMOM module. The similar configuration of

coupled Eulerian (Princeton Ocean Model) and Lagrangian (STRiPE) models was recently used for simulation of delivery and fate of fluvial water and terrigenous sediments discharged by the Peinan River at the south-eastern coast of the island of Taiwan under freshet and typhoon discharge conditions (Osadchiev et al., 2016) and to study dynamical features of the Zhuoshui and Wu river plumes located on the western coast of Taiwan (Korotenko et al., 2014).

## 5.1 INMOM module

An outer Eulerian model is the Institute of Numerical Mathematics Ocean Model (INMOM), a σ-coordinate ocean circulation model based on the primitive equations of ocean hydrothermodynamics with the Boussinesq and hydrostatic approximations. The global version of INMOM was used as the oceanic component of the IPCC climate model INMCM in the framework of the Coupled Model Intercomparison Project Phase 5 (Volodin et al., 2010), as well as for modelling of the Arctic Ocean (Johnson et al., 2012) in the frame of the Arctic Ocean Model Intercomparison Project, South Ocean (Downes

et al., 2015; Farneti et al., 2015) and the North Atlantic Ocean (Danabasoglu et al., 2016) in the frame of the Coordinated Ocean-ice Reference Experiments.

In this study we used a regional version of INMOM developed for the Black Sea. It employs polar coordinates for the horizontal dimensions and σ-coordinates for the vertical dimension. INMOM model domain covers the whole Black Sea basin to avoid open boundaries (Fig. 5). It is divided into 642 × 715 horizontal grid points (radius and azimuth), the pole is

located at the RCBS (40.205° E, 43.84° N). INMOM horizontal spatial resolution increases from 200 m at the north-eastern part of the Black Sea to 4.5 km at its south-western part. Thus, INMOM, on the one hand, reproduces mesoscale circulation of the whole Black Sea with moderate spatial resolution and, on the other hand, provides high spatial resolution at the study region, which is a serious advantage of INMOM as compared to other Eulerian ocean models in context of this work. Several works used this regional version of INMOM for studying aspects of the general circulation of the Black Sea (Zalesny et al.,

2012; 2013) and the coastal circulation at RCBS (Diansky et al., 2013; Zalesny et al., 2016a; 2016b).

A Laplacian operator along isopycnical surfaces is used for parameterization of the lateral diffusion of salinity and temperature, while a bi-Laplacian operator along σ-surface is used for the lateral viscosity on momentum (Volodin et al., 2010). The vertical coordinate is represented by 20 σ-levels with irregular vertical spacing to provide higher resolution near





the surface. The vertical viscosity and diffusion coefficients were parameterized as a function of the Richardson number according to the Pacanowsky and Philander scheme (Pacanowsky and Philander, 1981). The minimum water depth in the model domain was set equal to 3.5 m.

Surface and lateral boundary conditions for the INMOM model runs were set in the following manner. No heat and salt flux as well as no flow across the bottom and lateral solid boundaries were prescribed. No-slip and quadratic bottom friction conditions with drag coefficient $C_d$ equal to $2.5 \cdot 10^{-3}$ were applied to the lateral and bottom boundaries respectively. The initial temperature and salinity distributions were set according to the three-dimensional monthly mean climatic fields of the Black Sea with 25-50 km spatial resolution.

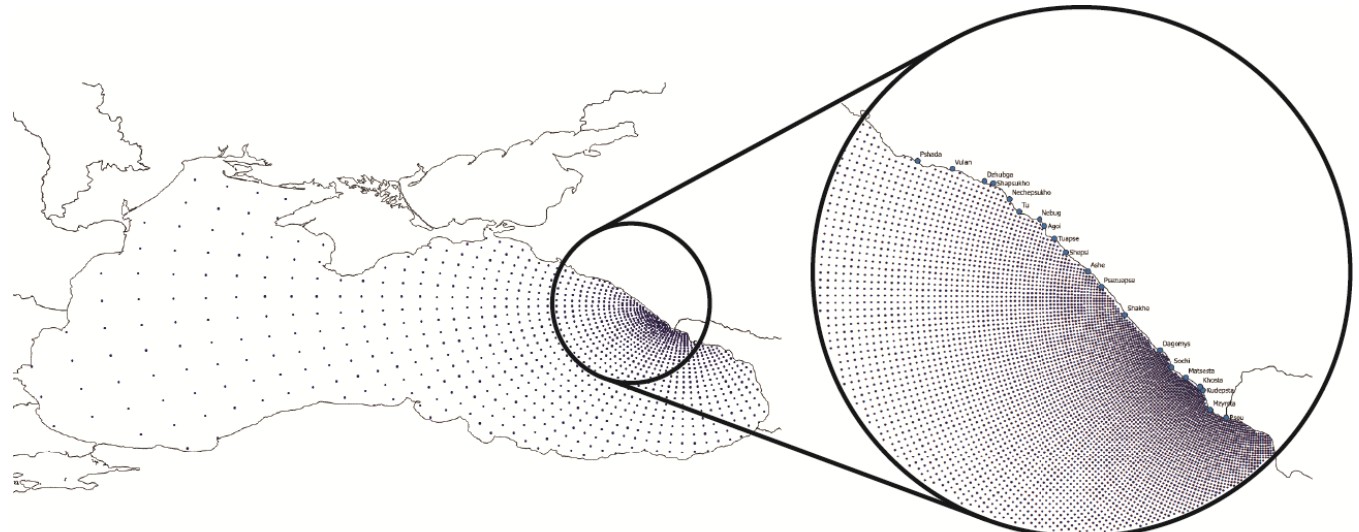

**Figure 5: INMOM model domain at the Black Sea (every 15th grid point is shown) (left) and at RCBS (every 4th grid point is shown) with indication of location of the considered river estuaries (right).**

The surface turbulent fluxes are calculated using the bulk formulae. The INMOM module was forced with climatological runoff of the 14 biggest rivers inflowing into the Black Sea and water transport through the Kerch and Bosporus straits, which were set according to the data provided in Jaoshvili (2002). The atmospheric forcing was adopted from the WRF data set. Tidal currents at the Black Sea are low and do not significantly influence general sea circulation; therefore, tidal forcing was not considered in numerical modelling. The INMOM module was initialized using the monthly averaged climatic temperature and salinity fields with 25-50 km spatial resolution as the initial background conditions and provided the ambient velocity fields for the STRiPE and sediment transport modules. Due to inaccuracy of available precipitation and river discharge, the corrections of the simulated SST and sea surface salinity (SSS) fields were applied using daily satellite SST products provided in the framework of the CMEMS project and monthly averaged climatic fields of SSS of the Black Sea. The model time step was set equal to 90 s.



## 5.2 STRiPE module

INMOM provides boundary conditions for the inner Surface-Trapped River Plume Evolution model (STRiPE) which was developed for high resolution simulation of sub-mesoscale dynamics of buoyant river plumes. STRiPE is a Lagrangian model which reproduces river plume as a set of imaginary "particles", i.e., homogeneous elementary water columns of reduced salinity extending from the surface down to the boundary between the plume and the underlying sea water. These particles are discharged into the sea from river estuaries, their initial velocity and height are governed by the river discharge rate and river estuary depth respectively. The subsequent motion of individual particles and their mixing with the underlying sea water are tracked by the model. The overall set of particles represents the river plume, and hence the temporal evolution of the plume structure is obtained. The main advantage of STRiPE lies in its ability to provide realistic results at relatively low computational cost as compared to Eulerian models. STRiPE was applied for studying plumes formed by small and medium size rivers of RCBS by Osadchiev and Zavialov (2013) and Osadchiev (2015).

Horizontal turbulent diffusivity used in the STRiPE module is parameterized by the Smagorinsky diffusion formula (Smagorinsky, 1963). Vertical mixing with the ambient seawater is parametrized by the salinity diffusion equation: $\frac{\partial S}{\partial t} = K_v \frac{\partial^2 S}{\partial z^2}$, where $S$ is the salinity, and $K_v$ is the vertical diffusion coefficient, which is parameterized via the Richardson number $Ri$ and scaling coefficient $C_v$ as given by Large (1994): $K_v = C_v (1 - \min(1, Ri^2))^3$. The detailed description of the STRiPE is given in Osadchiev and Zavialov (2013).

The STRiPE model domain covers the area from 38.0 to 40.2° E and from 43.1 to 44.6° N. The STRiPE module was forced by the WRF wind forcing, coastal circulation provided by the INMOM module, and the river discharge prescribed according to the data described in Section 4.2. The initial inflow velocities of the rivers $w_i$ were calculated using the Manning formula $w_i = 0.3 \left(\frac{Q_i}{L_i}\right)^{0.4}$, where $Q_i$ is the river discharge, $L_i$ is the width of the river mouth. The river and ambient sea water densities were prescribed as 1000 kg m$^{-3}$ and 1017 kg m$^{-3}$ respectively. The model time step was set equal to 90 s.

## 5.3 Sediment transport module

The transport and settling of fine suspended sediments discharged from the river mouth was simulated using a Lagrangian particle-tracking module. Both horizontal and vertical movements of a sediment particle were calculated using a combination of a deterministic component defined by motion of ambient water and sinking of a particle under the gravity force and a stochastic random-walk scheme that reproduces influence of small-scale turbulent mixing. Particles are initially released from the river mouth and their horizontal transport is determined by internal dynamics of a river plume simulated by the STRiPE module. After the sediment particle settles beneath the plume its movement is governed by the ambient coastal circulation, reproduced by the INMOM module. TSM concentrations at the river water were prescribed according to the data described in Section 4.2, while the sediment grain distribution was set based on the granulometric analysis of suspended sediments at water samples collected at the considered rivers during field surveys.



In this study we focus on relatively small particles with diameter less than $10^{-4}$ m; therefore, gravity-induced vertical motion is determined by a Stokes' law, and particle settling velocity $v_s$ is calculated according the well-known formula (Stokes, 1901): $v_s = \frac{gd^2(\rho_s - \rho_w)}{18\mu\rho_w}$, where $g$ is the gravitational acceleration, $d$ is the diameter of a sediment particle, $\rho_s$ is the density of a sediment particle prescribed equal to 2300 kg m$^{-3}$, $\rho_w$ is the density of water, $\mu$ is the dynamic water viscosity prescribed

equal to $10^{-3}$ Pa s$^{-1}$.

The total vertical displacement of the particle caused by sinking under gravity force, vertical circulation of ambient water, and turbulent mixing was parametrized using the following equation (Hunter et al., 1993; Visser, 1997; Ross and Sharples, 2004), which represents features of spatially non-uniform turbulent mixing: $\Delta z = \left(w_s + \frac{\partial K_v}{\partial z}\right)\Delta t + \sqrt{\frac{2}{3}K_v\left(\frac{1}{2}\frac{\partial K_v}{\partial z}\Delta t\right)\Delta t}\,\xi$, where $\Delta z$ is the vertical particle displacement, $w_s$ is the Stokes' settling velocity of a particle, $K_v$ is the vertical eddy

diffusivity coefficient, $\Delta t$ is the time step, $\xi$ is a random process with standard normal distribution (zero mean and unity variance) produced by a random number generator.

## 6 Results and discussion

The numerical simulations were organized in the following manner. First, the INMOM, STRiPE, and sediment transport modules were validated against in situ and satellite data. Then the numerical model was applied to simulate dynamics of the

river plumes formed by the 20 biggest rivers of RCBS ant the related delivery and fate of fine terrigenous sediments during two trial periods, 25 May – 4 July 2011 and 6-19 September 2011. The crucial reason of selection of these trial periods was their good coverage by optical remote sensing. Cloudless coastal area of RCBS was visible at EnviSat MERIS satellite imagery at 17 out of 41 days of the first period and 9 out of 14 days of the second period.

The first model run (25 May – 4 July 2011, "real" mode) represented typical spring-summer freshet discharge of large rivers

of RCBS and relatively low discharge of small rivers. The second model run (6-19 September 2011, "real" mode) simulated typical autumn drought discharge conditions for all rivers of the study region. Four intense rain events (25-27 May, 30 May, 21-23 June, and 27-30 June) during the first trial period and one (9-12 September) during the second trial period caused a considerable increase of discharge of small rivers and the subsequent decrease to average seasonal values as it was discussed in Section 1. The third and the fourth model runs ("averaged" modes) reconstructed spread of the river plumes during the

trial periods under seasonal discharge conditions without flash floods. For this purpose the "real" river discharge values were averaged over a period of 10 days. As a result the total discharge volumes of fluvial water did not change, but their daily hydrographs were significantly modified during the flooding periods, i.e., the discharge peaks were significantly smoothed.

We also simulated transport and settling of river-borne suspended sediments discharged during the first trial period under the "real" and "averaged" conditions and reconstructed distributions of fine sediments deposited to the seabed in the study area.

Thus, the numerical simulations in "averaged" mode reproduced delivery and fate of fluvial water and suspended matter under average climatic discharge conditions, i.e., in absence of flash floods, while "real" model runs combined both average



climatic and flooding discharge conditions. Therefore, comparing the model outputs in "real" and "average" modes we can distinguish the effect of short-term, but intense flash flooding events on transport, mixing, and settling of fluvial water and terrigenous sediments at the study area. In particular, based on the analysis and comparison of the obtained results of the "real" and "averaged" model runs, we reconstructed transport patterns of river-borne suspended sediments for normal and flash flooding discharge conditions during freshet and drought seasons.

## 6.1 Model validation

The INMOM module simulated general circulation of the Black Sea during the period from January to September 2011. The main large scale and mesoscale circulation features described in Section 2.3 were adequately reproduced by the numerical modelling, including the RC, the three quasi-stationary cyclonic gyres at the central divergence zone, and the four quasi-stationary anticyclonic gyres at the eastern part of the Black Sea near Sebastopol, Kerch, Batumi, and Gelendzhik. These features are visible in Fig. 6, which shows the modelled Black Sea current field at the depth of 10 m projected on a uniform grid with 10 km spatial resolution and averaged for June 2011. The model also reproduced seasonal variations of sea surface circulation, in particular, winter-spring intensification of RC, meandering of the main flow of RC caused by baroclinic instability, and formation of multiple NAE during summer at the eastern part of the Black Sea (Oguz et al, 1992, 1993; Titov, 2002). The simulated main flow of RC shifts offshore (approximately 45 km) in June and its speed decreases to 20 cm/s (Fig. 6) which is in a good agreement with in situ measurements described by Titov (2002).

The nested model was applied to simulate dynamics of river plumes of RCBS during the trial periods, 25 May – 4 July 2011 and 6-19 September 2011. The majority of days within these periods were characterized by typical spring-summer freshet and autumn drought conditions. The model at every time step outputs surface distributions of salinity and suspended sediments, thus the "real" mode simulations were validated against in situ salinity (Fig. 7) and TSM (Fig. 8) measurements, as well as satellite-derived TSM distribution maps (Fig. 9-10).







**Figure 6: The general surface circulation scheme of the Black Sea (1 – mean position of the cyclonic RC; 2 – quasi-stationary cyclonic gyres; 3 – Sebastopol, 4 – Kerch, 5 – Gelendzhik, 6 – Batumi quasi-stationary anticyclonic gyres) (top) and the INMOM modelled Black Sea circulation at the depth of 10 m averaged for June 2011 (bottom).**



Low computational cost of the Lagrangian model STRiPE enabled us to reproduce dynamics of the individual river plumes with high spatial resolution. In particular, the model adequately reproduced submesoscale variability of the Mzymta plume observed during the field survey on 28-30 May 2011, which is illustrated by Fig. 7-8. The spring freshet discharge of the Mzymta River was equal to 90-100 $m^3 s^{-1}$ during the end of May 2011. On 28 and 30 May 2011 the Mzymta plume was

stretched in a southern direction from the river mouth in response to moderate northerly and north-westerly wind. However, a significantly different wind forcing and, therefore, plume position and shape were observed on 29 May 2011. The Mzymta plume was arrested near the estuary by a slight onshore wind, which caused increase of its area and salinity anomaly near the estuary. The average positions of the simulated plume corresponding to 12:00-18:00 LT on 28-30 May 2011 illustrate this dramatic displacement and show good agreement with the salinity maps of the region obtained from the continuous CTD

measurements in the surface layer (Fig. 8).

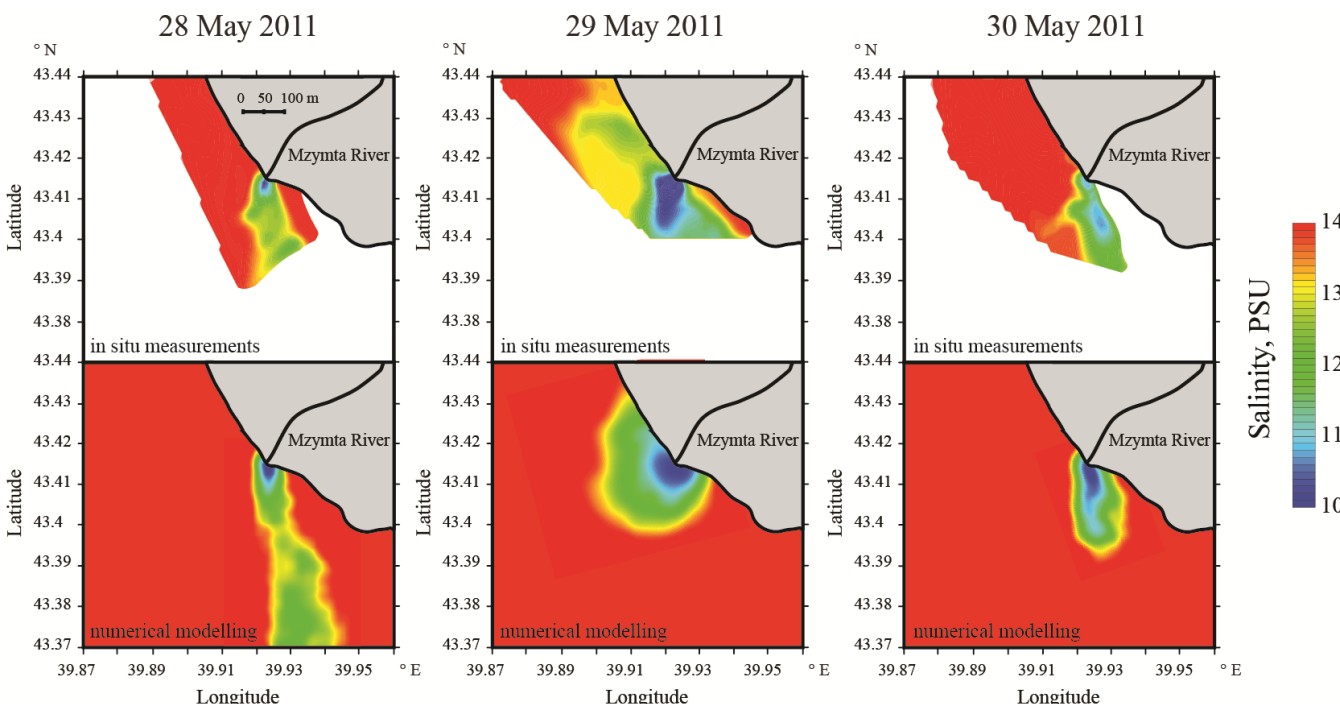

**Figure 7: Measured (top) and modelled (bottom) salinity distributions near the Mzymta River estuary on 28-30 May 2011.**

The simulated sediment distribution within the Mzymta plume was also validated against the in situ data. For this purpose we used continuous measurements of TSM concentration in the surface layer performed by the Ultraviolet Fluorescent LiDAR during the field survey and compared them with the simulated concentrations of suspended sediments. Both field data and numerical modeling showed that surface waters with elevated turbidity corresponded to the position of the plume




during the whole simulation period (Fig. 8). Sediment concentration in the study region increased on 29 May in response to slight wind forcing, which was reproduced by numerical modeling.

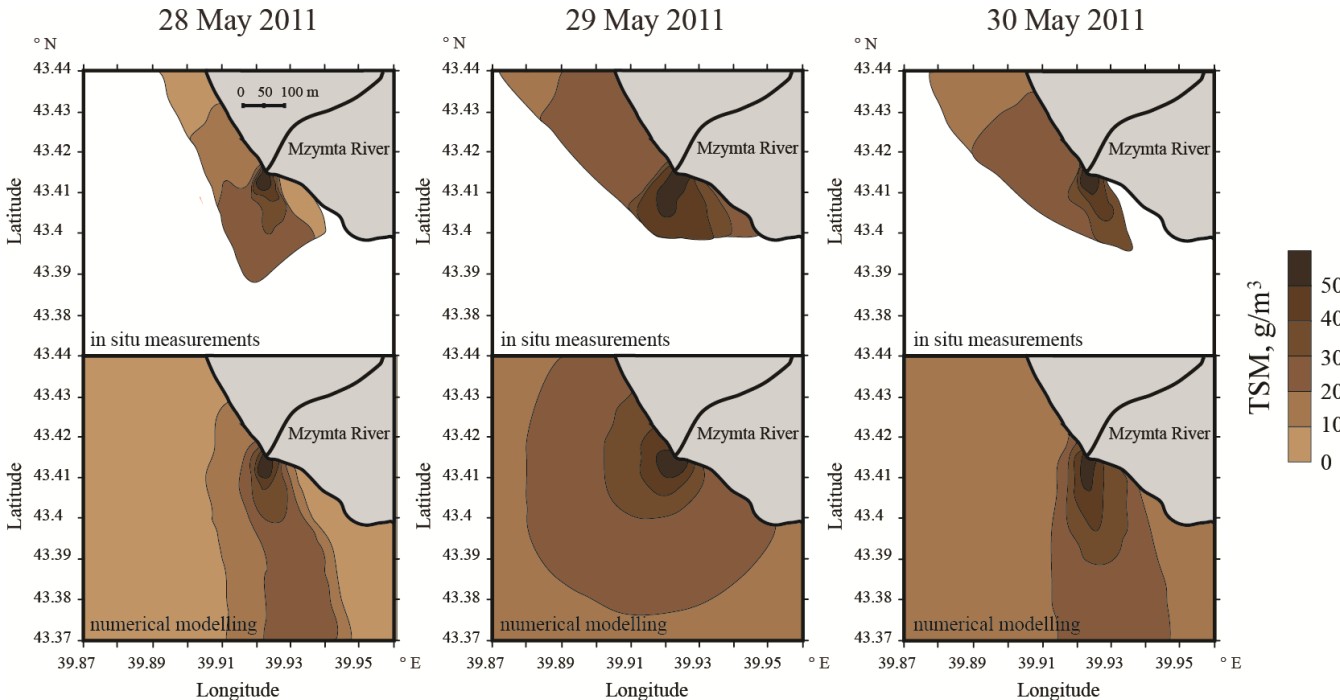

**Figure 8: Measured (top) and modelled (bottom) TSM distributions near the Mzymta River estuary on 28-30 May 2011.**

**6.2 Flash floods under "real" discharge conditions**

The typical example of a dramatic increase of areas of the buoyant plumes formed by the rivers of RCBS along a large segment of the coast in response to an active precipitation event and their subsequent decrease was observed on 26 June – 4 July 2011 (Fig. 9-10). On 26 June 2011 the Mzymta River was the only significant source of fluvial water and terrigenous

sediments at the coastal area of RCBS, hereinafter, referred as the point-source discharge pattern. The Mzymta plume was stretched along the shore in a northern direction, its area was approximately 30 km². Plumes formed by the Psou, Sochi, and other smaller rivers of RCBS on 26 June 2011 were considerably smaller than the Mzyta plume. Their spatial scales did not exceed 1-2 km and their impact on coastal water quality was negligible. Suspended sediment discharge rate from the Mzymta River was 4.8 kg s⁻¹, while total sediment discharge from the other modelled rivers of RCBS was almost twice less

(2.5 kg s⁻¹). River-borne terrigenous sediments were transported northward by the Mzymta plume, the coarse fraction was deposited mainly along the shore in proximity of the Mzymta estuary, while the fine fraction was transported offshore by the ambient coastal circulation.



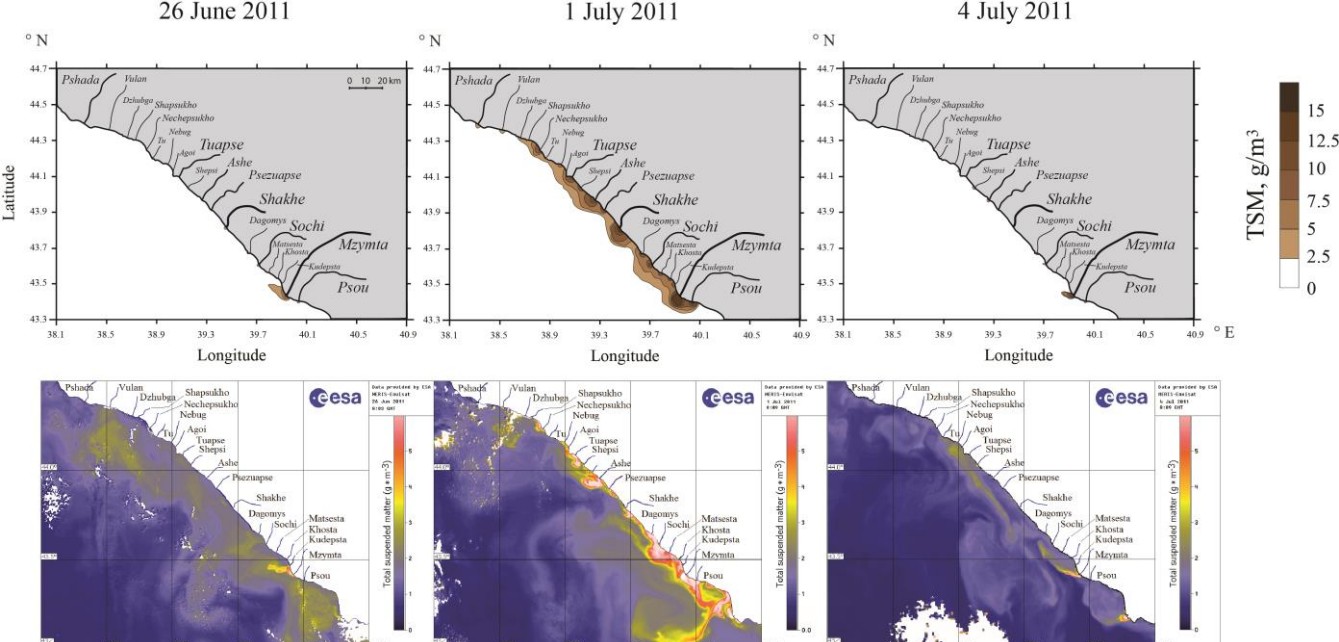

**Figure 9: Modelled (top) and satellite derived (bottom) TSM distributions at RCBS under the "real" mode on 26 June, 1 July, and 4 July 2011.**

Heavy rains, which lasted during 27-30 June 2011, affected the majority of rivers of RCBS and caused fast and substantial rise of continental discharge of fluvial water and suspended sediments, especially from the small rivers of the study region (Fig. 4). On 29 June individual river plumes coalesced into a continuous coastal stripe of freshened (14-17 PSU) and turbid (2.5-7.5 g m$^{-3}$) water between the Dzhubga and Psou rivers (Fig. 9). This stripe was formed by multiple sources located along the shore, hereinafter, referred as the line-source discharge pattern. Its alongshore length exceeded 150 km, its cross-

shore width was 5-15 km, and its depth was up to 5 m. The discharge of suspended sediments increased by one order of magnitude and was equal to 25-50 kg s$^{-1}$ for the Mzymta River, 10-25 kg s$^{-1}$ for the Psou, Shakhe, and Sochi rivers, 1-5 kg s$^{-1}$ for the small rivers of RCBS. As a result, large volumes of coarse sediments were deposited near multiple river estuaries along the shore. The intense line-source discharge also caused formation of strong alongshore geostrophic current (up to 40 cm s$^{-1}$) within the freshened stripe, which is typical for far-field regions of river plumes (e.g., Garvine, 1987; O'Donnell,

1990; Fong and Geyer 2002; Horner-Devine et al. 2006). This surface current caused intense transport of fine terrigenous sediments in the direction of Kelvin wave propagation, i.e., in a north-western direction.

After the end of the peak discharge period the line-source discharge pattern switched to the point-source pattern, the freshened stripe steadily diminished during several days and eventually dissipated on 3 July. Distributions of salinity and TSM at the study area on 4 July were similar to those observed before the flooding event (Fig. 9). The only large river plume

(approximately 20 km$^2$) was formed by the Mzymta River, while the areas of the other river plumes decreased to their average seasonal scales.



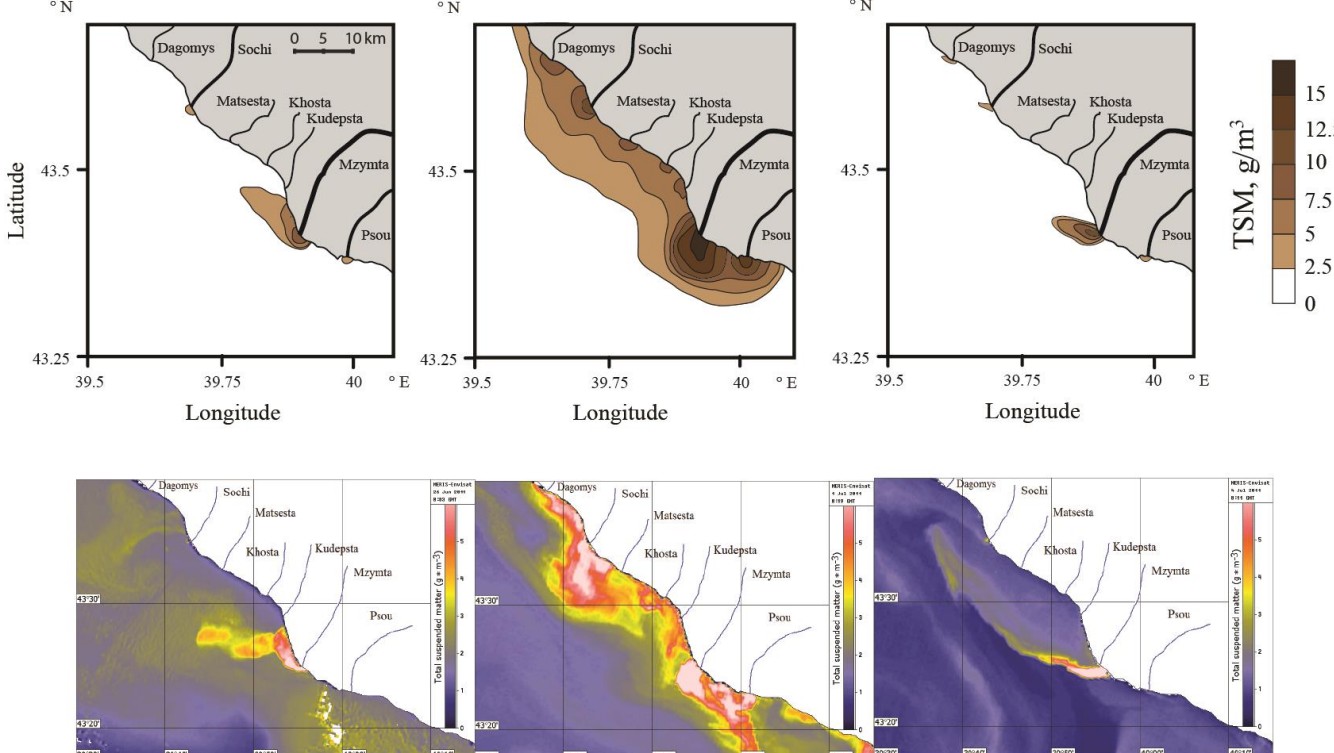

**Figure 10: Modelled (top) and satellite derived (bottom) TSM distributions at the southern part of the study region under the "real" mode on 26 June, 1 July, and 4 July 2011.**

Influences of three other flash flooding events (25-27 May, 30 May, and 21-23 June 2011) on coastal waters during the first trial period were also reconstructed by the numerical modelling. The heavy rain event, which took place at the southern part of the study region during 21-23 June 2011, caused formation of the line-source discharge pattern between the Shakhe and Mzymta rivers. Discharge of fluvial water and suspended sediments from the  individual rivers increased by one (from 1-10 $m^3 s^{-1}$ to 20-100 $m^3 s^{-1}$) and two (from 0.01-0.1 $kg s^{-1}$ to 2.5-25 $m^3 s^{-1}$) orders of magnitude respectively. The freshened and turbid stripe between the Shakhe and Mzymta rivers (60 km long) was observed during 21-24 June and dissipated on 25 June. Alongshore geostrophic current was also wide (up to 15 km), but less intense (up to 25 $cm s^{-1}$), as compared to that observed in the end of June and the beginning of July 2011.

Two less intense and shorter flash flooding events, as compared to those described above, were observed on 25-27 May and 30 May 2011 at the southern part of RCBS between the Shakhe and Psou rivers. They caused moderate increase of the river discharges (up to 25 $m^3 s^{-1}$), except the Shakhe (50 $m^3 s^{-1}$ on 25-27 May) and Psou (85 $m^3 s^{-1}$ on 30-31 May) rivers. As a result the areas of the river plumes were not large enough to collide, so these moderate flooding events did not cause formation of the freshened alongshore stripe.



Influence of a rain-induced flooding event on coastal waters was also studied during a draught period in September 2011. The discharge rates of fluvial water and suspended sediments from the Mzymta, Psou, Shakhe, and Sochi rivers were equal to 6-15 $m^3$ $s^{-1}$ and 0.05-0.4 kg $s^{-1}$, while the total discharge rates from the small rivers were 10-15 $m^3$ $s^{-1}$ and less than 0.1 kg $s^{-1}$. The spatial scales and salinity anomalies of the river plumes did not exceed 1 km and 2 psu. The rain-induced flooding

event, which was formed during 10-13 September 2011, caused increase of discharge of the small rivers by one order of magnitude, while discharge of the big rivers rose twofold or less. As a result, the discharge of terrigenous sediments from the small rivers increased by two orders of magnitude (1-8 kg $s^{-1}$) and exceeded sediment discharge from the big rivers (0.5-1 kg $s^{-1}$). However, total volume of continental discharge during 10-13 September 2011 was less than during 27-30 June 2011. A stripe of freshened and turbid water was formed between the Pshada and the Psou rivers, but it was not continuous and had

several gaps 5-10 km long. The maximal width (5 km) and depth (3 m) of the stripe were also significantly less than those observed in the end of June 2011. Nevertheless, the geostrophic current (10-20 cm $s^{-1}$) within this stripe caused north-eastward transport of fine terrigenous sediments during 4 days till the end of the flash flood on 14 September 2011, when the freshened stripe dissipated.

## 6.3 "Averaged" mode numerical experiments

The "averaged" mode experiments simulated delivery and fate of fluvial water and suspended sediments of RCBS during the trial periods in absence of flooding events. Averaging of the discharge hydrographs over a period of 10 days significantly influenced the small rivers by reducing the peak discharge values. As a result the variability of the total discharge of the small rivers during the first trial period changed from 21-224 $m^3$ $s^{-1}$ in the "real" mode to 40-108 $m^3$ $s^{-1}$ in the "averaged" mode. Averaging of the discharge rates of the large rivers of RCBS modified them less considerably, e.g., variability of the

Mzymta discharge changed from 47-139 $m^3$ $s^{-1}$ to 54-91 $m^3$ $s^{-1}$.

Under the "averaged" discharge conditions the Mzymta and Psou rivers were the only significant sources of fluvial water and terrigenous sediments during the whole first trial period. Spatial scales of the Mzymta and Psou plumes exceeded 10 km, while the other rivers did not form plumes greater than 1-2 $km^2$ even during the period of their maximal discharge on 1 July 2011 (Fig. 11). Total sediment discharge rate of the Mzymta and Psou rivers was 8-28 kg $s^{-1}$, while the total discharge rate of

all other rivers did not exceed 3.5 kg $s^{-1}$. Thus, the point-source discharge pattern was observed during the whole first trial period. Coarse fractions of suspended sediments were deposited mainly in proximity of the Mzymta and Psou estuaries, while fine fractions were transported offshore to the deep ocean by the ambient coastal circulation.

During the second trial period spatial scales of all river plumes of RCBS under the "averaged" discharge conditions were less than 1 km. As a result, the point-source discharge pattern was observed during the whole trial period. Due to absence of

flooding events characterized by intense sediment runoff, the daily sediment discharge from RCBS under the "averaged" discharge conditions was much smaller than under the "real" discharge conditions and did not exceed 3 kg $s^{-1}$. Thus, the rate of sediment load to the seabed was negligible under these conditions at the whole study area.



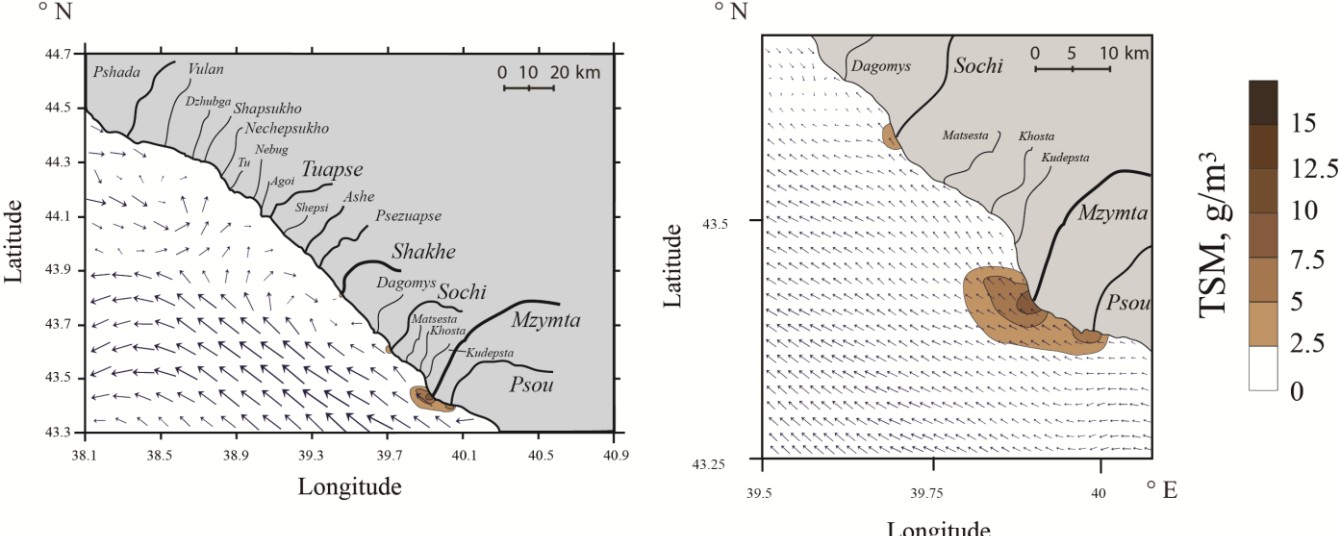

**Figure 11: Modelled TSM distribution at RCBS (left) and its southern part (right) under the "averaged" mode on 1 July 2011.**

## 6.4 Sediment transport and deposition

Besides salinity and TSM distributions at the study area, the nested model simulated transport and settling of river-borne
terrigenous sediments discharged from the rivers of RCBS during the first trial period under the "real" and "averaged"
discharge conditions. The obtained distributions of terrigenous sediments deposited to the seabed at the study area are
presented in Fig. 12.

Total discharge volumes from the individual rivers during the first trial period were the same for both simulation modes;
however, daily discharge rates were strongly different especially during flash flooding periods. Due to non-linearity of the
dependence of sediment discharge rate on water discharge rate, the total sediment flux from the modelled rivers during 41
days of the first trial period under the "real" mode ($8.5 \cdot 10^7$ kg) was greater by a quarter than under the "averaged" mode
($6.7 \cdot 10^9$ kg).

The resulting deposit patterns obtained for two modes also were significantly different. Under the "averaged" discharge
conditions the majority of river-borne sediments was discharged from the Mzymta, Psou, Sochi, Shakh, and Tuapse rivers
and was deposited in proximity of their estuaries. The region of the most active sediment load (0.2-0.5 kg m$^{-2}$) provided by
the Mzymta and Psou rivers was located at the southern part of the study region, its alongshore extent and area were
approximately 20 km and 60 km$^2$. The area of the region adjacent to the Mzymta estuary, where sediment load exceeded 0.4
kg m$^{-2}$, was approximately 5 km$^2$.

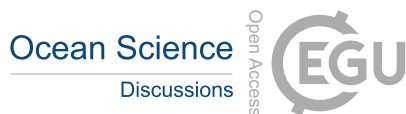

**Figure 12. Simulated distribution of terrigenous sediments discharged from the rivers of the study area during 25 May – 4 July 2011 and deposited to the seabed at RCBS (top) and its southern part (bottom) under the "real" (left) and "averaged" (right) modes.**

As in the "averaged" mode run, the Mzymta and Psou rivers were significant sources of suspended sediments at the study region under the "real" discharge conditions. However, the flash flooding events, reproduced in the "real" mode resulted in significant sediment runoff from the other rivers of RCBS. Moreover, formation of the strong alongshore geostrophic currents of turbid and freshened water on 21-24 June and 29 June – 2 July 2011 resulted in intense transport of sediments along the shore in a north-western direction. In particular, a large volume of sediments discharged from the large southern rivers (Psou, Mzytma, Sochi, and Shakhe) were transported several tens of kilometers far from their sources and settled on the seabed at the large segments of the shelf area. As a result, sediment load exceeded 0.1 kg m$^{-2}$ along almost the whole



shore between the Psou and Dzhubga rivers. The total alongshore length of the coast influenced by active sedimentation was approximately 150 km, which is significantly greater than the corresponding length (35 km) observed under the "averaged" discharge conditions.

## 7 Summary and conclusions

This study is focused on the influence of the river plumes of RCBS on delivery and fate of fluvial water and terrigenous sediments under average climatic and flooding discharge conditions. For this purpose, we used a nested combination of the Institute of Numerical Mathematics Ocean Model (INMOM) and the Surface-Trapped River Plume Evolution model (STRiPE). The Eulerian model INMOM reproduced general ocean circulation at the north-eastern part of the Black Sea and provided boundary conditions for the Lagrangian model STRiPE, which was used for simulating dynamics of river plumes.

The model was validated against in situ measurements and satellite imagery. In order to study influence of flash floods on delivery and fate of fluvial water and terrigenous sediments during freshet and drought seasons we considered two trial periods, 25 May – 4 July 2011 and 6-19 September 2011. Based on in situ data, satellite imagery, and numerical modelling, we reconstructed the daily values of fluvial water and terrigenous sediments discharged during the trial periods fromthe 20 biggest rivers of the study region. Then for both trial periods we simulated spread of the buoyant plumes in two modes, first,

using the reconstructed discharge data ("real" mode) and, second, excluding flash flooding periods by averaging the discharge data over a period of 10 days ("averaged" mode).

  The numerical experiments showed that short term rain-induced flooding events significantly influence sediment transport and deposition patterns at RCBS. Under average climatic discharge conditions total runoff of fluvial water and terrigenous sediments is dominated by several largest rivers of the study area. Water and sediment yield from the small rivers is low;

therefore, their plumes have small spatial scales, high dissipation rate, and their impact on coastal water quality is negligible. As a result, continental discharge significantly influences water quality and induces intense sediment load only in proximity of the estuaries of the large rivers.

  Active precipitation events can cause rapid and substantial increase of water and sediment discharge from the small rivers of RCBS and can induce formation of flash flooding conditions at the long segments of seashore during both freshet and

drought seasons. Under these conditions the areas of the river plumes dramatically increase and the individual plumes can collide and coalesce with neighbouring ones. As a result the related change of the discharge pattern from point-source to line-source can cause formation of one or several alongshore stripes of freshened and turbid water up to 200 km long and greatly transform transport pathways of river-borne suspended and dissolved matter at the study region. These stripes, first, influence dynamics of the river plumes by decreasing mixing intensity between the plumes and the ambient water, thus,

increasing the spatial scales of the river plumes. Second, alongshore geostrophic currents of turbid and freshened water, which are formed within these stripes, induce intense transport of sediments in a north-western direction and their settling



along the shore. This process significantly influences coastal water quality and causes active sediment load at large segments of narrow shelf of RCBS as compared to average climatic discharge conditions.

RCBS is a densely populated area (approximately 1.1 million people) and the most important recreational area of Russia which is visited by more than 10 million people annually. 95% of the residential and visitor population is located at the narrow coastal area less than 10 km far from the sea shore; therefore, correct evaluation of the influence of the small rivers on coastal water quality is extremely important for this region. As it was shown above, flash flooding events influence discharge from the small rivers much more considerably as compared to the large rivers. Therefore annual distribution of precipitation, in particular, frequency, intensity, and time spacing of extreme precipitation events significantly affects land-ocean fluxes of freshwater, dissolved and suspended matter. Steady increase of temperature (1.2-1.4 ºC) and annual precipitation volume (32-164 mm) at the study region registered in 1971-2010 was accompanied by increase of number of heavy rains. As a result both the number of extreme river floods and the maximal annual discharge volume also show growing tendency (Alexeevsky et al., 2016). This fact confirms the importance of study of the small rivers of RCBS and their role in local land-ocean fluxes of freshwater, terrigenous sediments, nutrients, and pollutants.

Influence of small rivers on delivery and fate of fluvial water, dissolved and suspended matter can be significant for many world regions, which have similar configuration of rivers inflowing to sea, i.e., close spacing of river mouths, steep and small river basins, frequent flash flooding events. This configuration is typical for mountainous coastal areas with humid climate, e.g., Chile (Saldias et al., 2012; 2016), south-western coast of USA (Mertes and Warrick, 2001; Nezlin et al., 2008), south-western coast of New Zealand, south-eastern coast of Papua New Guinea, etc. Thus, the results obtained in this study related to influence of point-source and line-source discharge conditions on transport patterns of terrigenous sediments at RCBS could be applied for other coastal areas such as those, described above.

**Acknowledgements**

The authors are grateful to many colleagues from Shirshov Oceanology Institute for valuable support during the fieldwork and wish to thank Peter Zavialov, Nikolay Diansky, Dmitry Soloviev, Vadim Pelevin, Boris Konovalov, Alevtina Alyukaeva, and Vladimir Belokopytov whose data were used in this study. The authors wish to thank the European Space Agency and the Copernicus Marine Environment Monitoring Service for the provided satellite data and the Federal Service for Hydrometeorology and Environmental Monitoring of Russia for the provided river discharge data. This research was funded by the Russian Scientific Foundation (research project 14-17-00382).



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
