# Peer review of "Small river plumes off the north-eastern coast of the Black Sea under average climatic and flooding discharge conditions"

_Ocean Science, 2017_

## Referee Comment (RC1) · Anonymous Referee #1 · 7 Mar 2017

Review of Small river plumes off the north-eastern coast of the Black Sea. . .. By Osadchiev and Koshenko

Overall: This is a generally well-written paper on an interesting and novel topic. The effects of small rivers have been noted for some time (e.g. Milliman & Syvitski, 1992). This new manuscript employs a well-validated numerical modelling approach to illustrate the importance of event-driven discharges from small rivers, how they affect the coastal system very differently to predictions based on mean discharges, how small rivers can alter the coastal system dramatically compared to just considering the dominant river sources, and including the effects of both the salinity of the coastal water and the delivery and transport of fine sediments. The results are novel and very interesting.

General suggestions: While the focus is on salinity and suspended sediments, what are the consequences for the nutrients coming down the rivers? Global, catchment-model-based estimates of annual river discharges and nutrient loads (e.g. the NEWS 2 database, see Mayorga et al., Env. Modelling Software, 2010) which, for the majority of rivers around the world, represent the best information currently available. These models tend to provide discharges and loads in an annual mean sense – so what are the implications of this new work, for both suspended material and also for nutrients? It struck me that one big difference between the few large rivers and the many smaller ones would be river length and, possibly, the catchment type being drained. It would be worth some sensible speculation in the discussion to consider the implications of the work for the use of such global databases.

There is some discussion (page 34, lines 9-13 or so) on trends in event-driven discharges. Are these climate-change driven or local regional natural variability? Either way, it could be clarified, but also the possible climate-driven changes to more extreme events generally could be drawn out more here.

More detailed suggestions: The validation of the satellite-derived suspended material data with river discharge could be more robustly demonstrated, e.g. plots of ln(C) vs ln(Q) for large and small rivers, and a demonstration that the correlation coefficients are significant.

Some of the details in the model configuration could perhaps be edited out, as there are sufficient published studies that have already set up the model that can provide these.

Section 6.1 seemed a bit out of place. Either the validation of the model should be part of the methods, or it should at least occur at the start of the results section.

The continuous along-shore low salinity plume (line 10, page 18) is noted as being 5 – 15 km in width. How does this compare to the local internal Rossby radius?

Page 21, lines 11-12. Check the sediment load numbers – they are different by 2 orders of magnitude, and they don't seem consistent with the statement immediately following about the "real" system being 25% greater.

There is heavy use of abbreviations in the manuscript (e.g. RCBS, GCR) which gets confusing at times. My preference is to avoid abbreviations unless they are very widely accepted – the text flows better without the reader having to keep reminding themselves what an abbreviation stands for.

The manuscript will need some careful checking for editing/clarification of English – though it is generally very well written.

---

## Referee Comment (RC2) · Anonymous Referee #2 · 11 Apr 2017

Summary: The authors present a comprehensive study concerning the impact of sediment and freshwater delivery from river input on freshening, plumes and sediment transport in the Russian Coast of the Black Sea. The approach used in the modelling study is to consider the influence of river input on the spread of buoyant plumes using two different scenarios: including periods of flash flooding and averaging conditions over a 10-day period. The study employs a suite of models to reproduce the discharge from the 20 largest rivers along the Russian Black Sea coast. The three models implemented are the INMOM hydrodynamic model for the entire Black Sea, with polar coordinates in the horizontal such that resolution is as high as 200m in the

area of interest. The STRiPE model, a high resolution Lagrangian model for simulation of buoyant plumes is implemented for the river discharges. A Lagrangian particle–tracking model is employed to simulate transport and settling of fine suspended sediments discharged from river mouths. The models are well validated from available data and understanding. The INMOM model reproduced the major large and meso scale circulation patterns and features. The STRiPE reproduced the submesoscale variability of individual river plumes, and in particular the Mzymta plume, the largest of the 20 rivers included in the study. Sediment transport and distribution was validated with Ultraviolet Fluorescent LiDAR which measured TSM concentration in the upper layer. The main result is that realistic rain-induced (flash) flooding events influence considerably sediment transport and deposition along the Russian Black Sea coast, while under averaged conditions river runoff and sediment deposition and transport is dominated by the largest rivers.

Comments: This is a broad and interesting study that presents a number of novel concepts, ideas and tools, and deserves to be published in OS subject to a revision. The main issue I have concerning the manuscript is the language and style, the English. This is the weakest part of the paper. It is often awkwardly written and needs to be improved. I understand that English is not the first language of the authors. However, the good news is it can be improved. The authors would need to go through the paper thoroughly, preferably with a scientist whose first language is English. This will constitute technical corrections and minor revision. Section 5 p11 The INMOM model was initialized from climatology. Is this satisfactory? Was there any spin-up involved or considered? Please discuss. P11 line 17- 24 please provide references From p.10 on: the INMOM module should be the INMOM model, same for STRiPE module and Sediment Transport module Section 6.1 please provide references for model validation, e.g. for general Black Sea circulation and smaller eddies. p.13, line 4-5 units incorrect $\mu$âŇŁkgmˆ(-1) sˆ(-1) âŇŃ Fig. 7 – 10 captions and text. Please clarify/state what is top and bottom. From p.18 on stripe(s) should be strip(s)

---

## Author Comment (AC1) · 4 May 2017

C: While the focus is on salinity and suspended sediments, what are the consequences for the nutrients coming down the rivers? Global, catchment-model-based estimates of annual river discharges and nutrient loads (e.g. the NEWS 2 database, see Mayorga et al., Env. Modelling Software, 2010) which, for the majority of rivers around the world, represent the best information currently available. These models tend to provide discharges and loads in an annual mean sense – so what are the implications of this new work, for both suspended material and also for nutrients? It struck me that one big

difference between the few large rivers and the many smaller ones would be river length and, possibly, the catchment type being drained. It would be worth some sensible speculation in the discussion to consider the implications of the work for the use of such global databases.

R: Thank you for these valuable suggestions. Spatial resolutions of basins and river networks used in these models are not enough to reproduce small rivers as those addressed in this work. As a result, these models neglect the role of small rivers in nutrient and sediment loads. However, the extreme flooding events can, first, significantly influence freshwater, nutrient, and sediment land-ocean fluxes at least at a synoptic time scale, and, second, have significantly different patterns of coastal transport of dissolved and suspended water constituents as it was shown in this work for RCBS. Thus, this study can be used for estimation of the role of small rivers in nutrient and sediment loads at certain world coastal areas with similar river discharge configurations which are neglected by the global catchment-based models. The related discussion was added to the manuscript.

C: There is some discussion (page 34, lines 9-13 or so) on trends in event-driven discharges. Are these climate-change driven or local regional natural variability? Either way, it could be clarified, but also the possible climate-driven changes to more extreme events generally could be drawn out more here.

R: Thank you for this comment. These trends are climate-change driven which was clarified in the manuscript. Also according to your suggestion we added a discussion about the climate-driven changes in frequency and duration of the intense and extreme precipitation events in the study region.

C: The validation of the satellite-derived suspended material data with river discharge could be more robustly demonstrated, e.g. plots of ln(C) vs ln(Q) for large and small rivers, and a demonstration that the correlation coefficients are significant.

R: According to your recommendations we added the plots of ln(C) and ln(Q) for the

large and small rivers of the study region (Fig. 5) and the relevant discussion to Section 4 to demonstrate that the correlation coefficients are significant.

C: Some of the details in the model configuration could perhaps be edited out, as there are sufficient published studies that have already set up the model that can provide these. Section 6.1 seemed a bit out of place. Either the validation of the model should be part of the methods, or it should at least occur at the start of the results section.

R: We agree that some details of the model configuration given in Section 5 can be edited out, especially concerning the STRiPE and sediment transport models. According to your suggestions we significantly shortened the model description in Section 5 and moved the model validation from Section 6.1 to Section 5.

C: The continuous along-shore low salinity plume (line 10, page 18) is noted as being 5 – 15 km in width. How does this compare to the local internal Rossby radius?

R: Many thanks for this comment. The cross-shore extent if this strip was ∼5 km, however, in proximity of the estuaries of the large rivers of RCBS it increased up to 10-15 km. This is consistent with the value of the local internal Rossby radius equal to 5 km. This clarification was added to the text.

C: Page 21, lines 11-12. Check the sediment load numbers – they are different by 2 orders of magnitude, and they don't seem consistent with the statement immediately following about the "real" system being 25% greater.

R: Thank you for this point, corrected.

C: There is heavy use of abbreviations in the manuscript (e.g. RCBS, GCR) which gets confusing at times. My preference is to avoid abbreviations unless they are very widely accepted – the text flows better without the reader having to keep reminding themselves what an abbreviation stands for.

R: We agree that the multiple abbreviations used in the manuscript are confusing and can be mostly avoided. We replaced all abbreviations except RCBS (Russian coast of

the Black Sea), which is the longest and the mostly often used (58 times) and several well-known and widely used like TSM, CDOM, SST, SSS, WRF, STRiPE, INMOM.

C: The manuscript will need some careful checking for editing/clarification of English – though it is generally very well written.

R: Thank you for this comment. The revised manuscript was proofread by an expert English speaker, which significantly improved its language and style.

On behalf of all authors, Alexander Osadchiev

Please also note the supplement to this comment:
http://www.ocean-sci-discuss.net/os-2017-1/os-2017-1-AC1-supplement.pdf

---

## Author Comment (AC2) · 4 May 2017

C: The main issue I have concerning the manuscript is the language and style, the English. This is the weakest part of the paper. It is often awkwardly written and needs to be improved. I understand that English is not the first language of the authors. However, the good news is it can be improved. The authors would need to go through the paper thoroughly, preferably with a scientist whose first language is English. This will constitute technical corrections and minor revision.

R: Thank you for this comment. The revised manuscript was proofread by an expert

[Figure]

English speaker, which significantly improved its language and style.

C: Section 5 p11 The INMOM model was initialized from climatology. Is this satisfactory? Was there any spin-up involved or considered? Please discuss.

R: Thank you for this point. Usage of climatological values of temperature and salinity from databases and assuming the zero velocity field at the start is a common approach for initializing numerical models of ocean. We believe that the climatological temperature and salinity fields used in this study model are one of the best available datasets for the Black Sea. In this work we considered the upper ocean dynamics for a regional model of the Black Sea with daily SST and monthly SSS data assimilation. As a result, the model reached a statistically steady state after a relatively short spin-up period of 5 months (January – May 2011). The related discussion was added to the manuscript.

C: P11 line 17- 24 please provide references

R: We added references about tidal currents in the Black Sea, as well as for surface salinity and temperature data sets used for initialization and correction of the INMOM model.

C: From p.10 on: the INMOM module should be the INMOM model, same for STRiPE module and Sediment Transport module

R: Corrected

C: Section 6.1 please provide references for model validation, e.g. for general Black Sea circulation and smaller eddies.

R: We added additional references about large scale and mesoscale circulation features of the Black Sea reproduced by the INMOM model.

C: .13, line 4-5 units incorrect ÌĄFig. 7 – 10 captions and text. Please clarify/state what is top and bottom. From p.18 on stripe(s) should be strip(s)

R: Corrected

On behalf of all authors, Alexander Osadchiev

Please also note the supplement to this comment:
http://www.ocean-sci-discuss.net/os-2017-1/os-2017-1-AC2-supplement.pdf

———————————————

[Figure]

**Supplement:**

**AUTHORS' REPLY TO THE REVIEWERS' COMMENTS AND THE LIST OF CHANGES MADE IN THE REVISED MANUSCRIPT OS-2017-1**

We appreciate the reviewers' comments that served to improve the article. The following changes were made in the manuscript in response to the suggestions made by the referees.

**Comments:**

*The main issue I have concerning the manuscript is the language and style, the English. This is the weakest part of the paper. It is often awkwardly written and needs to be improved. I understand that English is not the first language of the authors. However, the good news is it can be improved. The authors would need to go through the paper thoroughly, preferably with a scientist whose first language is English. This will constitute technical corrections and minor revision.*

Thank you for this comment. The revised manuscript was proofread by an expert English speaker, which significantly improved its language and style.

*Section 5 p11 The INMOM model was initialized from climatology. Is this satisfactory? Was there any spin-up involved or considered? Please discuss.*

Thank you for this point. Usage of climatological values of temperature and salinity from databases and assuming the zero velocity field at the start is a common approach for initializing numerical models of ocean. We believe that the climatological temperature and salinity fields used in this study model are one of the best available datasets for the Black Sea. In this work we considered the upper ocean dynamics for a regional model of the Black Sea with daily SST and monthly SSS data assimilation. As a result, the model reached a statistically steady state after a relatively short spin-up period of 5 months (January – May 2011). The related discussion was added to the manuscript.

*P11 line 17- 24 please provide references*

We added references about tidal currents in the Black Sea, as well as for surface salinity and temperature data sets used for initialization and correction of the INMOM model.

*From p.10 on: the INMOM module should be the INMOM model, same for STRiPE module and Sediment Transport module*

Corrected

*Section 6.1 please provide references for model validation, e.g. for general Black Sea circulation and smaller eddies.*

We added additional references about large scale and mesoscale circulation features of the Black Sea reproduced by the INMOM model.

*p.13, line 4-5 units incorrect*

*Fig. 7 – 10 captions and text. Please clarify/state what is top and bottom.*

*From p.18 on stripe(s) should be strip(s)*

Corrected